# Micro-Structures Produced by Crystal Growth from Located Nuclei and Their Transfer Aiming at Functional Surfaces

**Nobuyuki Moronuki [1],\* and Renato Serizawa [2]**

[1] Faculty of Systems Design, Tokyo Metropolitan University, Tokyo 191-0065, Japan
[2] Graduate School of Systems Design, Tokyo Metropolitan University, Tokyo 191-0065, Japan; serizawa-renato@ed.tmu.ac.jp
\* Correspondence: moronuki@tmu.ac.jp

**Abstract:** Hydrothermal processes can produce regular micro-/nano-structures easily; but their placement or position is difficult to control, and the obtainable structures tend to be random. For controlling the crystal growth, two types of definite and regular structures were obtained. The first ones were ZnO urchin-like structures synthesized from located ZnO particles as the nuclei. These structures were found to work as gas sensors utilizing a wide surface area. The second one was a vertically aligned $TiO_2$ nanorod array synthesized on a fluorine-doped tin oxide substrate that has a similar lattice constant to rutile $TiO_2$. Super-hydrophobicity after ultraviolet irradiation was then examined. Finally, the synthesized $TiO_2$ array was peeled off and transferred onto a resin sheet. We determined that the substrate could be subjected to repeated hydrothermal synthesis, thereby demonstrating the reusability of the substrate. These results demonstrate the applicability of these processes for industrial applications.

**Keywords:** self-organizing process; microstructure; self-assembly; crystal growth; nuclei

## 1. Introduction

Self-organizing or bottom-up processes such as self-assembly or any kind of crystallization process can produce various regular and functional structures [1], while top-down processes like cutting or lithography serve as dominant manufacturing processes.

Zinc oxide (ZnO) is a stable material that has semiconductor characteristics [2]. Aligned structures have been produced and applied to light emitting diodes [3], and flower-like microstructures have been produced and applied to sensors [4]. In addition to thermal evaporation processes [5], the hydrothermal process is a simple process that can produce regular and intricate structures, such as urchin-like [6] or aligned structures [7]. One of the problems with this process is the randomness of the structure's position. By applying the self-assembly of fine particles [8] as the nuclei of the subsequent hydrothermal process, the structural position can be determined, and, by applying a wettability pattern on the substrate, the position of the assembly can be precisely controlled [9]. However, the number of such proposals is limited.

Titanium dioxide ($TiO_2$) is another stable and functional material. Using the hydrothermal process, this material can also be synthesized. By choosing fluorine-doped tin oxide (FTO) as a substrate, vertically aligned nanorods can be obtained due to the similarity of the lattice constants of rutile $TiO_2$ and FTO [10]. Applications were discussed for the improvement of solar cell efficiency [11] and energy harvesting with aligned $BaTiO_3$ nanorods, which were produced by a subsequent hydrothermal process from aligned $TiO_2$ and offered appropriate piezoelectric characteristics [12].

The FTO substrate is a conductive glass and is convenient when the structure is applied to electrical devices because the substrate can be used as one of the electrodes. However, there are other requirements for transferring these structures to another substrate because the thinness of the structure at a micrometer level makes the structure flexible.

In cases with both ZnO and TiO$_2$, by arranging the nuclei of the crystal growth or substrate, definite and regular structures are expected to emerge. However, relevant methods have not been established, thus limiting applications. We previously sought to apply the self-assembly of fine particles [9] and transfer them to another substrate using ultraviolet curing resin [13]. By applying these processes, the TiO$_2$ nanorod array can be transferred to another substrate. However, there are no such trials thus far. Therefore, this study proposes a method that utilizes hydrothermal synthesis and the transfer of the synthesized thin layer to widen the applications of these functional materials.

## 2. Proposed Procedures

Figure 1 shows the aims of this study. The hydrothermal process is a simple process that proceeds in a sealed container at an elevated temperature. By controlling the initial conditions of crystal growth and utilizing the principle of selective crystal growth at the required location, fine and regular structures can be obtained [14,15]. One such example is the ZnO urchin-like structures that result from multi-directional crystal growth with fine particle as the nucleus, and another is an array of aligned titanium dioxide nanorods caused by directional crystal growth on a special substrate, as shown in Figure 1a.

Figure 1b shows the schemes of the nuclei's arrangement. The self-assembly of fine particles can be achieved by drawing a substrate at a constant speed $V$ from a suspension in which the particles are dispersed [9]. Ideally, monolayered assembly of the packed particles can be obtained. The principle of this process involves a combination of capillary force, which draws the suspension onto a substrate, and meniscus attraction during evaporation of the water, which facilitates the packed assembly of the particles. Silica particles of a 1 μm diameter have often been assembled [9,13]. To obtain uniform assembly, the following conditions are important: (1) The water contact angle of the substrate should be smaller than about ten degrees; (2) the aggregation of particles due to electrostatic forces should be avoided by modifying the pH, if necessary; (3) the drawing speed $V$ and its angle $\alpha$ should be determined based on preliminary experiments.

Figure 1c shows a cross-sectional view of the transfer scheme. Liquid UV resin is placed onto the structure and penetrates into the space between the rods. After UV irradiation, the cured resin was peeled from the substrate together with the structure. At this stage, the affinity of the resin with the structure is important because a bad affinity will result in a mis-transfer.

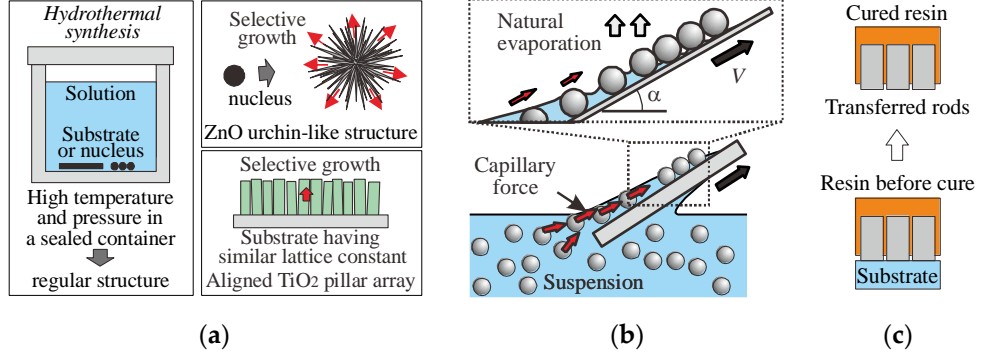

(**a**)  (**b**)  (**c**)

**Figure 1.** Schematics showing the aims of this study. (**a**) Hydrothermal synthesis is a simple process in a sealed container at an elevated temperature. Depending on the crystal structure and controlled nucleation, fine and regular structures can be obtained; (**b**) self-assembly of fine particles via the dip-coating process. Ideally, a monolayered packed structure will be obtained; (**c**) transfer of the synthesized structure with resin.

## 3. Verification

### 3.1. ZnO Urchin-Like Structures Aiming at Gas Sensors

Figure 2a shows the principles of urchin-like production. ZnO has a hexagonal close-packed crystal structure, and its crystal growth proceeds selectively along the *c* axis in the figure. Ideally, a spherical particle will have many crystal facets. Thus, crystal growth will proceed in a random direction at the beginning, but the differences in crystal growth rates will result in a profile featuring the fastest crystal growth (i.e., urchin-like structures) [14].

Figure 2b shows the self-assembly results of the ZnO particles with diameters $\phi$ 100 nm, where the substrate was drawn up at 0.1 mm/s from the 1 wt% concentration suspension. In this case, the coverage of the particles on the substrate after dip-coating was low and insufficient. However, by repeating the coating process, the coverage of the particles increased, although the multilayered part can be observed. In this case, the substrate was covered with a thin layer of poly-dimethylpolysiloxane (PDMS, Sylgard 184) whose flexibility helped to not drop off but hold the particles on the substrate during the subsequent hydrothermal process [16].

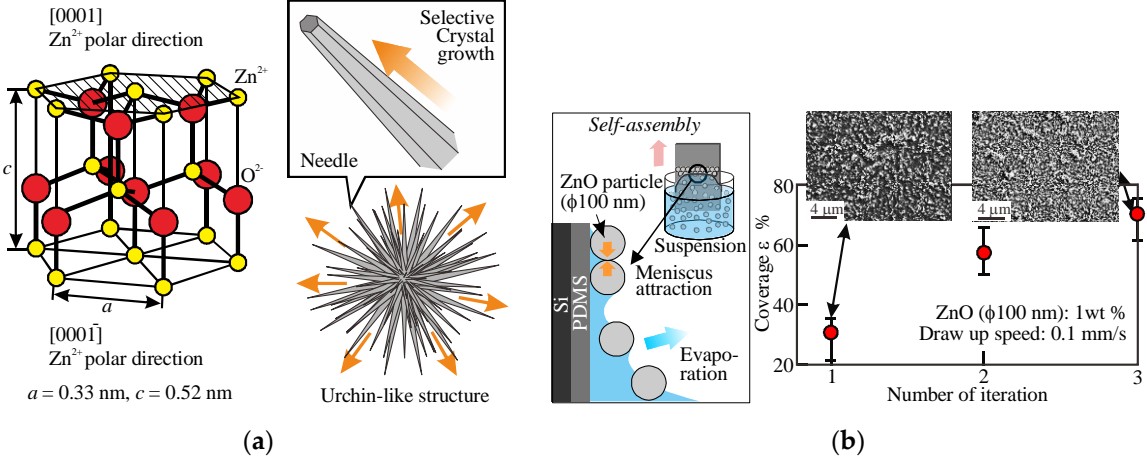

|  |  |
|:---:|:---:|
| (**a**) | (**b**) |

**Figure 2.** Principle of the urchin-like profile production and arrangement of the nuclei. (**a**) Hexagonal close-packed structure of the ZnO crystal and its selective growth from a nucleus in multiple directions results in an urchin-like profile; (**b**) the arrangement of ZnO nuclei using the dip-coating process. The coverage increased with the number of dip-coating iterations [15].

Table 1 summarizes the substrate preparation and hydrothermal process conditions for the ZnO urchin-like structure. In this case, the self-assembled ZnO particles play an important role in the arranged nuclei. Hydrothermal conditions are often difficult to determine; thus, preliminary experiments were carried out. Zinc nitrate and hexamethylenetetramine were the source materials, and ammonia was added to adjust the pH of the water-based solution. The pH values were changed between 8 and 11, and the appropriate value was set to 10, as the urchin diameter became small and particulate at other pH values.

**Table 1.** Experimental conditions of the ZnO urchin-like structures.

| Substrate, nuclei | Silicon wafer covered with polydimethylpolysiloxane (PDMS), ZnO particles ($\phi$ 100 nm) |
|:---:|:---:|
| Hydrothermal synthesis | Zn(NO$_3$)$_2$ 0.58 g, C$_6$H$_{12}$N$_4$ 0.29 g, water 20 mL, 90 °C, 9 h |

Figure 3 shows the results of the ZnO hydrothermal synthesis. The urchin-like structures were successfully synthesized, as observed from the scanning electron microscope (SEM) photos. The diameter increased with the reaction time, and the maximum diameter was $\phi$ 17 µm after a 24 h

reaction in this case. It is also found from the SEM photos that the distance between the urchins increased together with an increase in the diameter. The average distance between the urchins differed from the original distance between the particles and nuclei, and the distance increased with the urchin diameter and reaction time. These results suggest that the small urchins dissolved and were then synthesized as larger urchins. In the following, the detailed reaction process will be discussed.

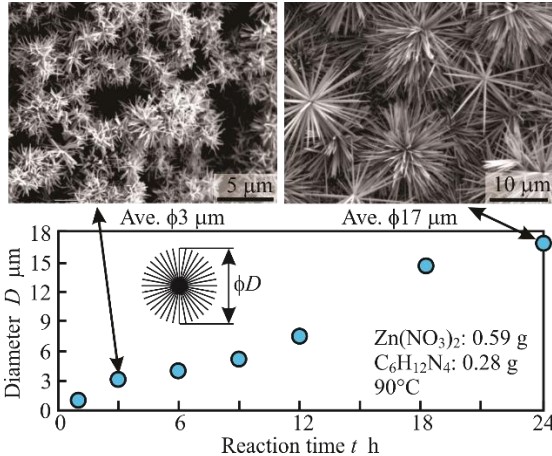

**Figure 3.** Results of the urchin-like structure production showing the relationship between the reaction time and urchin diameter together with SEM pictures. The diameter increased linearly under these conditions [15].

The reaction process was analyzed using Equations (1)–(6) [16]. Equation (6) suggests that the material synthesis reaction is not one-way but bi-directional. Thus, it is predicted that the material will dissolve along with the synthesis depending on the conditions. However, this is a hypothesis, and further elucidation is necessary.

$$C_6H_{12}N_4 + 6\,H_2O \rightarrow 6\,HCHO + 4\,NH_3 \tag{1}$$

$$NH_3 + H_2O \rightleftarrows NH_4{}^+ + OH^- \tag{2}$$

$$Zn^{2+} + NH_3 \rightarrow Zn(NH_3)_4{}^{2+} \tag{3}$$

$$Zn^{2+} + OH^- \rightarrow Zn(OH)_4{}^{2-} \tag{4}$$

$$Zn(NH_3)_4{}^{2+} + 2\,OH^- \rightleftarrows ZnO + 4\,NH_3 + H_2O \tag{5}$$

$$Zn(OH)_4{}^{2-} \rightleftarrows ZnO + H_2O + 2\,OH^- \tag{6}$$

Figure 4 shows the evaluation results of the ZnO structures as a gas sensor. Figure 4a shows a cross-sectional schematic of the sensor structure and the working principle with a picture. Urchin-like structures were sandwiched within the electrodes and kept at elevated temperatures up to 350 °C. Under this environment, the approaching ethanol gas molecules exchanged electrons at the ZnO structure surface, which caused a resistance change between the electrodes. The wide surface area of the urchin-like structure will increase the sensitivity of the gas sensor.

Figure 4b shows the performance of the gas sensor. The horizontal axis denotes the elapsed time before and after gas exposure, and the vertical axis denotes the impedance or resistance between the electrode measured with an LCR meter (3511-50, HIOKI E.E. Corp., Nagano, Japan). The experiments were carried out in a chamber 400 mm × 800 mm × 600 mm in size. The gas concentration was calibrated with a commercially available gas detector tube (112 L, GASTEC Corp., Kanagawa, Japan). This figure shows that the impedance of the urchin-like structure started with a lower value and then decreased down to a few megaohms after gas exposure, while another sensor made of ZnO particle structures provided a higher impedance value and slower response.

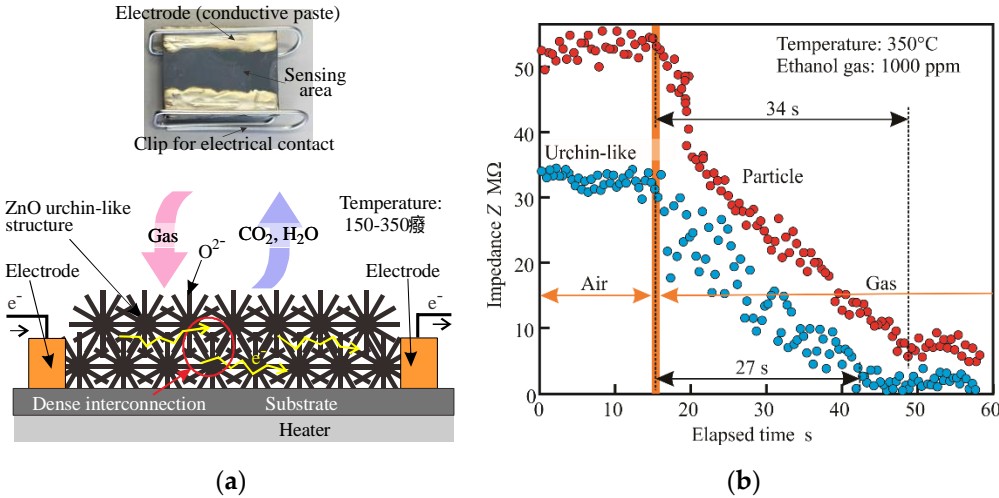

**Figure 4.** Results of the performance test as a gas sensor: (**a**) Schematics showing the sensor structure with a picture. The electron exchange between the gas molecules and ZnO urchins changed the resistance of the structure. The urchin structure helps the electron exchange due to its wide surface area; (**b**) experimental results of ethanol gas sensing showing a comparison between the assembled particles and urchins. The low resistance and quick change provided excellent sensor performance [15].

### 3.2. Vertically Aligned TiO₂ Nanorods for Superhydrophilicity

Titanium dioxide $TiO_2$ is a stable material with photocatalytic characteristics. By using these photocatalytic characteristics, a superhydrophilic surface can be obtained after ultraviolet irradiation, which has been already applied to the self-cleaning function used in building exteriors and antibacterial surfaces [17] for operating room interiors in hospitals. The photocatalytic characteristics of rutile-type $TiO_2$ are weaker than those of the anatase type $TiO_2$, but the rutile-type can be easily synthesized with the hydrothermal process.

Table 2 summarizes the substrate preparation and hydrothermal process conditions for $TiO_2$ nanorods. In this case, the lattice constant of the fluorine-doped tin oxide (FTO) substrate plays an important role in promoting epitaxial crystal growth. The hydrothermal conditions were determined based on previous studies [10]. The reaction process is simple, and a titanium tetrachloride was used as the source material.

**Table 2.** Experimental conditions for the $TiO_2$ nanorods.

| Substrate (nuclei) | Fluorine-doped tin oxide (FTO) |
|---|---|
| Hydrothermal synthesis | $H_2O$ 30 mL, HCl 60 mL, $TiCl_4$ 1 mL, 150 °C, 2–12 h |

Figure 5 shows the principles and some results of the $TiO_2$ nanorod synthesis. Figure 5a shows the principle of the aligned crystal growth. The lattice constant of the rutile $TiO_2$, $a = b = 0.473$ nm is similar to that of the fluorine-doped tin oxide (FTO) (0.473 nm). In this case, the crystal growth will align to the substrate, and then a crystallographically oriented structure can be obtained, just like epitaxial growth. However, a macroscopic crystal size is difficult to obtain, and the rod array is obtained via random nucleation [18].

In Figure 5a, the relationship between the reaction time and rod height is shown together with photographs of the sample. With an increase in reaction time, the rod height first increased linearly, but when the reaction time was longer than 8 h, the increase rate decreased. It can be found from the photographs that transparency decreased with an increase in thickness and that excess thickness caused a partial peel off. The cause of this peel off can be attributed to residual strain caused by the difference in the lattice constants between $TiO_2$ and FTO.

Figure 5b shows the results of the SEM observations. The observations were carried out from two directions: a top-view and a 45-degree oblique view. It can be found that a reaction time shorter than 4 h resulted in a uniformly aligned rod array. However, a reaction longer than 6 h resulted in the irregular formation of urchin-like structures on the top. Thus, the reaction time was set at 4 h in later experiments.

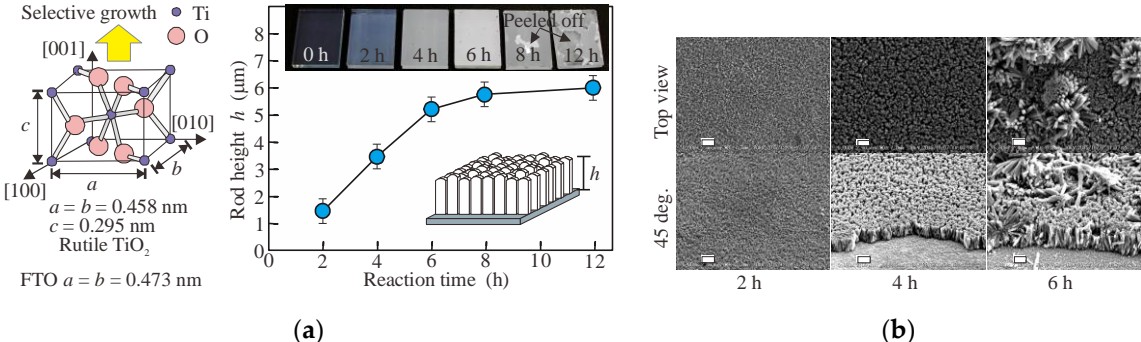

(a)                                                                      (b)

**Figure 5.** Principle and experimental results for titanium dioxide ($TiO_2$) nanorod synthesis: (**a**) Epitaxial crystal growth of $TiO_2$ on a fluorine-doped tin oxide (FTO) substrate requires similar lattice constants. The experimental results showed that the rod height increased linearly up to 6 µm, while further reactions caused peeling off from the rods; (**b**) results of SEM observation. Uniform aligned rod arrays were obtained when the reaction time was shorter than 4 h. Excess reaction time produced urchin-like structures on top.

The rod height distribution was examined using the analysis software of a white light interferometer (Talysurf CCI, AMETEK, PA, USA). Figure 6a shows the height distribution of the rods with good condition for the case of 4 h shown in Figure 5b, where the height distribution was less than 0.75 µm. The bad condition rods in the 6 h case (Figure 5b) showed a height distribution larger than 15 µm. Thus, it was found that appropriate hydrothermal synthesis conditions have a uniform height distribution.

Figure 6b shows the results of powder x-ray diffraction (XRD, Ultima IV, Rigaku Corp., Tokyo, Japan). As references, the FTO substrate and $TiO_2$ particles were analyzed together with the hydrothermal synthesis results. A strong peak was found at the (002) facet of $TiO_2$ for the synthesized results, while weak peaks of $SnO_2$ in the substrate were also observed. In the case of the $TiO_2$ particles, peaks of different facets of $TiO_2$ were observed, which is considered natural because the particles have random orientations on the substrate. In the case of the FTO substrate, no peaks in $TiO_2$ were found, while $SnO_2$ peaks were observed.

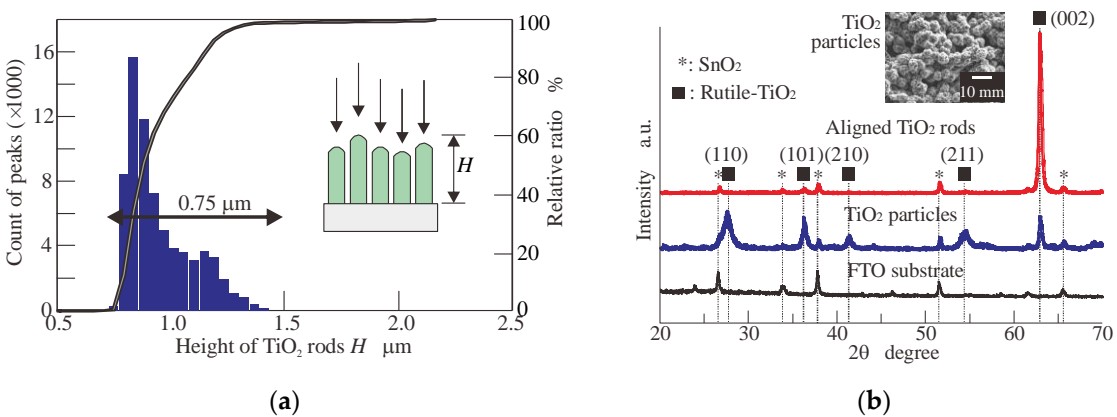

(a)                                                                      (b)

**Figure 6.** Characteristics of the synthesized $TiO_2$ rod array: (**a**) variations in the rod height of the good sample were less than 0.75 mm, while the bad sample presented variation of 15 µm; (**b**) a comparison of the x ray diffraction (XRD) results showing the crystallographically aligned structures of rutile $TiO_2$ [18].

Figure 7 shows the water contact angles on the TiO$_2$ rod array after ultraviolet (UV) irradiation. Figure 7a shows the effects of UV irradiation. The wavelength and the power of the UV light source were 365 nm and 38 mW/cm$^2$, respectively. With an increase in irradiation time, the contact angle decreased by less than 1 degree under the condition of superhydrophilicity due to the photocatalyst effect of TiO$_2$. As a reference, the contact angles of the FTO substrate are shown in the Figure 7a. Due to the decomposition of organic contamination on the surface, the contact angle decreased slightly, but the contact angles were larger than 35 degrees.

The measurement of the contact angle is difficult because the value changes over a time scale as long as a day. Figure 7b shows the change in water contact angle one week after UV irradiation. The synthesized TiO$_2$ rods maintained a low contact angle for one week after irradiation. The reference was a piranha-processed glass plate under the conditions shown in Table 3. The sample was kept in a desiccator in a clean room, but the contact angle increased over time which may be attributed to reconstitution of the end groups or organic contamination.

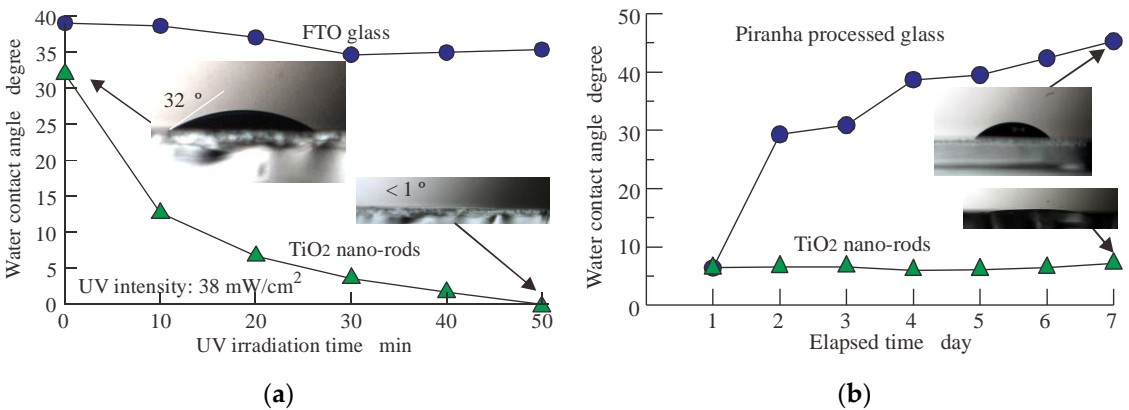

(**a**)  (**b**)

**Figure 7.** Characteristics of the wettability of the synthesized TiO$_2$ surface: (**a**) UV irradiation changed the surface to superhydrophilic; (**b**) change in the water contact angle for one week after irradiation.

**Table 3.** Conditions for the piranha process.

| | |
|---|---|
| Ultrasonic cleaning | Ethanol 5 min, Water 5 min |
| Piranha process | $H_2SO_4:H_2O_2 = 3:1$, 80 °C, 2 h |
| Rinse with pure water | 10 min |

*3.3. Material Transfer*

3.3.1. Transfer of the Assembled Particles

Before the transfer of the hydrothermally synthesized structures, transfer of the assembled particles with a resin layer was examined.

Figure 8 shows a scheme of the transfer of self-assembled particles and its results. Figure 8a shows a cross-sectional view of the particle transfer scheme. First, the particles are self-assembled on "Substrate 1" via dip-coating, as shown in Figure 1b. Ideally, monolayered assembly will be achieved, but structures often become multi-layered. Resin is then coated on "Substrate 2", half-cured to adjust the viscosity, and then situated upside-down on the assembly. By applying pressure between "Substrate 1" and "Substrate 2", ultraviolet light is irradiated through "Substrate 1"—in this case, to cure the resin. Afterward, two substrates are peeled off, and particles are transferred to "Substrate 2". By changing the half-curing conditions and contact pressure, the height $H$ can be controlled, though its range is limited. If the thickness of the resin can hold the multi-layered structure, the top of the resin structure will have the equivalent flatness of "Substrate 1".

Figure 8b shows SEM pictures of the transferred particles. The upper photo shows the wide-area observation result, indicating that a uniformly smooth surface was obtained. The lower pictures show the tops of the particles embedded in the resin. Here, both the monolayered and multilayered particles were properly held beneath the cured resin.

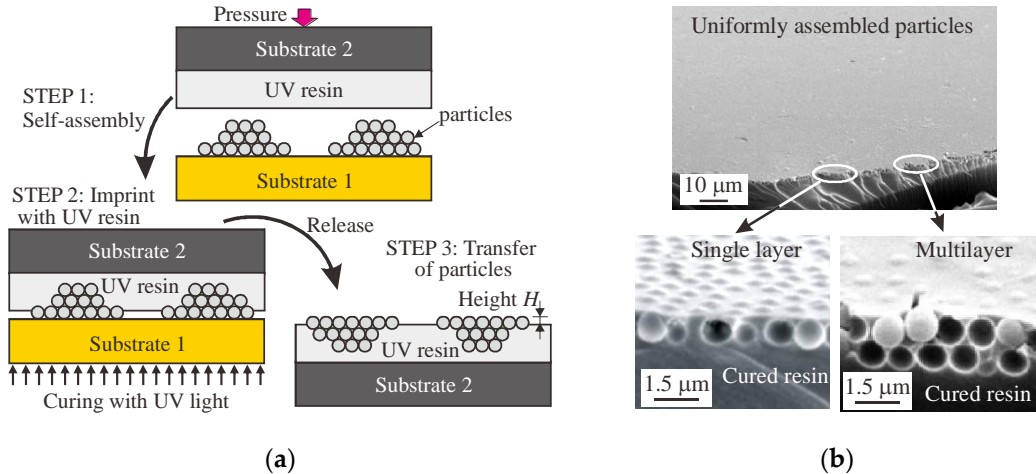

(**a**)	(**b**)

**Figure 8.** Scheme for the transfer of self-assembled particles and the results: (**a**) Schematic of the particle transfer using half-cured resin; (**b**) SEM photos of the particle transfer results. Assembled particles are uniformly transferred upside-down on the resin layer, holding both monolayered and multi-layered particles [13].

These results suggest that the nuclei of the thermal synthesis can be located in advance if the resin can hold the particles at an elevated temperature.

Two types of UV curing resins were applied to investigate the basic characteristics. One has organic material affinity with low viscosity of 8.8 mm$^2$/s (PAK-02, Toyo Gosei, Tokyo, Japan), while the other has inorganic material affinity and high viscosity of 56 mm$^2$/s (PAK-01, Toyo Gosei, Tokyo, Japan). Silica (SiO$_2$, inorganic) particles were self-assembled on a glass plate and then transferred onto a silicon substrate. In these experiments, the viscosity was modified from its original value via a half-curing process [19].

Figure 9 shows the case of resin with inorganic material affinity. With an increase in the contact or imprint pressure, the transfer depth increased. The transfer depth is defined as the ratio of the embedded depth $d$ to the particle diameter $D$, which was calculated by analyzing the top-view image of the inset SEM photos. In these cases, the top parts of the particles were exposed to the air, and the bottom parts were embedded in the resin.

When the resin with organic material affinity was applied, the particle profiles were transferred to the resin, resulting in a concave dimple array due to the resin's poorer affinity with the particles [19].

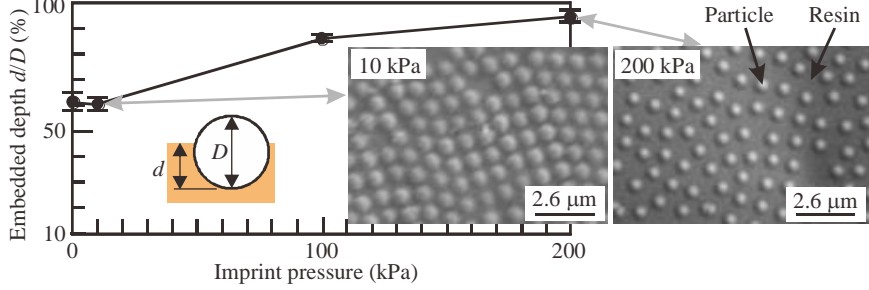

**Figure 9.** Effect of imprint pressure on the embedded depth of the assembled SiO$_2$ particles. The depth of embedment can be changed with the imprint pressure between the substrates.

### 3.3.2. Transfer of Nanorods

Hydrothermally produced ZnO urchin-like structures seem to be easily transferrable to other substrates because the contact area between the structures and the substrate is limited. On the other hand, $TiO_2$ nanorods are difficult to transfer because of the wide contact area between their structures and the substrate. For applications such as energy harvesting [20], the material transfer of nanorods structure to another substrate becomes important.

Using the UV resin, $TiO_2$ nanorods structures were transferred. The experimental conditions are shown in the Table 4, where the two types of resin described in the previous section were used. Figure 10a shows the principle of the transfer. The resin must penetrate into the narrow space between the rods via capillary force, and, after curing, it must have strong affinity with the rods to hold the rods during the transfer process. In Figure 10a, the resultant force of sharing force due to the affinity force $F_a$ and stiction force at the top of the rod $F_t$ must be larger than the bonding force $F_b$ between the rods and the substrate. If this condition cannot be satisfied, the rods will not be transferred and remain on the FTO substrate.

Figure 10b shows the results of the SEM observations after the transfer. The number in the lower-right in parentheses shows the condition ID (Table 4). IDs (1) and (2) both show the good transferring results. The flat and relatively smooth top surfaces indicate that the resin reached the bottoms of the rods before the transfer. The top surface of ID (3) looks very different from the surfaces of ID (1) and (2). In this case, the resin had a high viscosity, as shown in Table 4, as well as inorganic material affinity. Here, the resin did not reach to the bottom of the rods, and the rods could not be transferred and remained on the FTO substrate; thus, deep dimples are observed on the top surface of the cured resin.

**Table 4.** Experimental conditions for $TiO_2$ nanorod transfer.

| Condition ID | Array Spacing | Resin | |
| :---: | :---: | :---: | :---: |
| | | Affinity | Viscosity ($mm^2$/s) |
| (1) | Sparse | Organic (PAK-02) | 8.6 |
| (2) | Dense | Organic (PAK-02) | 8.6 |
| (3) | Sparse | Inorganic (PAK-01) | 56 |

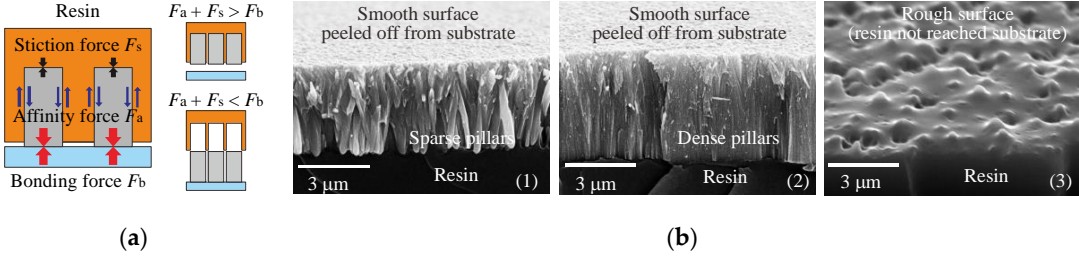

| | |
| :---: | :---: |
| (**a**) | (**b**) |

**Figure 10.** Transfer of the $TiO_2$ rod array, its principle and its results: (**a**) The resin viscosity should be low to penetrate between the rods and have affinity with the rods to hold them during the transfer; (**b**) SEM photos of the results. The left two cases show that the resin reached the bottom of the rods, while the right image shows a case where the resin did not reach the bottom of the rods. Numbers in parentheses show the condition ID in Table 4.

## 4. Discussion

Urchin-like structures can be applied to various kinds of chemical reactions in addition to gas sensors utilizing a wide surface area. Potential applications include reactors or catalysts. If necessary, the structures can be transferred to another substrate using the process described in Section 3.3. The advantage of the gathered arrangement is that electrical connections can be kept under substrate deflection because the complicated connecting points among the gathered urchins will maintain contact,

although their positions may change unless they are completely separated. Such characteristics were partially demonstrated in a previous study [15].

A superhydrophilic surface can be applied to self-cleaning functions, especially in a watery environment such as sanitary applications because the water layer thrusting between the surface and the adhered substance will allow the substance to float. The case of an oil-repellant surface was demonstrated in a previous study [21]. For aligned nanorods, different kinds of application can be considered, as follows.

Energy harvesting is expected to become an energy source for sensor networks to help create secure societies [20]. For example, environmental vibrational energy can be converted into electric energy. The energy level is limited but enough to activate a wireless sensor system at regular time intervals.

One of the traditional piezoelectric materials is lead zirconate titanate (PZT), and bulk structures are produced by sintering small grains of this substance. After sintering, a polarizing process is necessary because the original dipole directions of the grains are random. By applying a high electric field, the random directions of the grains can be aligned to produce a complete piezoelectric device. The aligned $TiO_2$ rod array can be converted into an aligned $BaTiO_3$ rod array by applying further hydrothermal processes. This aligned structure can eliminate the polarizing process along with the toxic lead elements. The attainable piezoelectric constant of $BaTiO_3$ is smaller than that of PZT; thus, good alignment and crystallinity are necessary to improve the energy conversion efficiency.

A thin film structure can permit elastic deformation like bending; thus, this structure can be used for wearable devices. The transfer processes described in the previous section will make this type of production feasible. In addition, if the vibrational mode can be analyzed in advance, patterned arrangements of the structure can be used to improve the generating efficiency. In a previous study, the patterned arrangement of particles was already demonstrated, although the experiments in the present paper are limited to a uniform structure.

## 5. Conclusions

A consistent process to produce regular structures with hydrothermal synthesis was proposed and demonstrated while partially referring previous studies. The results are summarized as follows:

- Zinc oxide (ZnO) urchin-like structures were successfully produced from ZnO particles located as nuclei, and their gas-sensor functionalities were demonstrated.
- Titanium dioxide ($TiO_2$)-aligned nanorods were produced on a fluorine-doped tin oxide (FTO) substrate, and their functionalities were examined. Then, the rods were transferred using an ultraviolet light curing resin layer.

**Author Contributions:** Conceptualization, N.M.; methodology, N.M.; investigation, R.S.; resources, R.S.; data curation, N.M.; writing—original draft preparation, N.M.; writing—review and editing, N.M.; visualization, N.M.; supervision, N.M.; project administration, N.M.; funding acquisition, N.M. All authors have read and agreed to the published version of the manuscript.

**Funding:** This research was partially funded by JSPS KAKENHI Grant Numbers JP21360065 and JP19K04108.

**Acknowledgments:** The authors express special thanks to the efforts of former students, Kogiso J., Takada K., Okubo Y., and Morita S., who conducted numerous experiments.

**Conflicts of Interest:** The authors declare no conflict of interest.

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
