# Peer review of "Micro-Structures Produced by Crystal Growth from Located Nuclei and Their Transfer Aiming at Functional Surfaces"

_jmmp, doi:10.3390/jmmp4040105_

Round 1
Reviewer 1 Report
The paper by Moronuki and Serizawa is a detailed description and characterization of micro-structures produced by crystal growth from located nuclei and their transfer aiming at functional surfaces. The manuscript is well- written, the results accurate and well presented. Before it can be accepted for publication on Journal of Manufacturing and Materials Processing, the following minor revisions are required.
1) Please, place Figure 3 before line 104.
2) Lines 104-105. The English style of this sentence should be improved.
3) Line 134. A reference should be given.
4) Line 138. Change “In the this case” into “In this case”.
5) An experimental set-up paragraph should be added, describing instruments (SEM; XRD measurements..), methods and data treatment used for the analysis.
6) Paragraph “4. Discussion” and Paragraph “5. Conclusions”. Even if the authors clarified the innovativeness of their study with respect to the state of art, they should better clarify, in both paragraphs, the possible future industrial applications of their research on these functional materials.
Author Response
Thank you for kindly review. Followings are the answer to the comments.
1) Please, place Figure 3 before line 104.
-> Revised according to the comment.
2) Lines 104-105. The English style of this sentence should be improved.
-> Revised according to the comment. Whole of the manuscript has been polished by a native speaker.
3) Line 134. A reference should be given.
-> Revised according to the comment.
4) Line 138. Change “In the this case” into “In this case”.
-> Revised according to the comment.
5) An experimental set-up paragraph should be added, describing instruments (SEM; XRD measurements..), methods and data treatment used for the analysis.
-> The authors once tried but it was difficult to gather as one paragraph because wide variety of experiments are described in the manuscript such as gas sensor test and water contact angle measurement. Thus, specific explanation is necessary in each part in the manuscript.
6) Paragraph “4. Discussion” and Paragraph “5. Conclusions”. Even if the authors clarified the innovativeness of their study with respect to the state of art, they should better clarify, in both paragraphs, the possible future industrial applications of their research on these functional materials.
-> Revised according to the comment. Potential applications are thoroughly included.
Reviewer 2 Report
This manuscript discusses the synthesis of nanorods, which are then studied for their functionality and are tranported van ultraviolet light curing resin layer. The manuscript is interesting, however, they are several additions and topics that need to be addressed before I would recommend to publish it.
Firstly, throughout the manuscript, the English language needs to be improved.
The following items need to be addressed:
Line 54 - The authors discuss controlled nucleation using hydrothermal synthesis. Often, it is thought that there is not a lot that can be controlled, as far as crystal growth is concerned, in hydrothermal synthesis. There is a reference (14), however, this reference appears to be a conference presentation. This is a pretty bold statement to control crystal growth hydrothermally, the authors should expand on how to do this a bit more.
Line 93 - A reference needs to be added after "Hydrothermal conditions are determined referring the previous researches"
Line 94 - what was the final pH after the solution was adjusted with ammonia? This is an important experimental detail.
Line 102-103 - The authors state that "This means that the small urchins dissolved and then snthesized as bigger urchins." Is there actual evidence of this? This should be described as a proposed theory since hydrothermal synthesis cannot be monitored and viewed in real-time.
Line 104 - referring to equations 1-6 - The authors should describe this ias a proposed mechanism. If there is evidence that this is certainly the mechanism - text should be added to address that. If they got this mechanism from somewhere else, it should be referenced.
Line 141 - In this case the pH was adjusted with HCl, however, the authors do not say what the pH was adjusted to, or how much HCl was added. The authors should say what pH the solution was adjusted to.
Line 142- referring to Table 2 - The authors show the chemicals used in this experiment in the table, however, they do not discuss ratios or amounts as they do in Table 1. This important experimental information should be added here.
Line 171 - The authors should specify that they are conducting Powder X-ray Diffraction.
Line 195- In the caption for Figure 7, the authors states that the "hydrophilicity was kept for a long time" and there is a reference present. I have two comments here, a "long time" is all relative, therefore, the authors should put a minimum amount of time here that the hydrophilicity was unaffected. Additionally, why is there a reference present here, when the hydrophility length of time is being discussed?
Author Response
Thank you for the kindly review. Followings are the answer to the comments. Regarding the English, the manuscript has been polished by a native speaker.
Line 54 - The authors discuss controlled nucleation using hydrothermal synthesis. Often, it is thought that there is not a lot that can be controlled, as far as crystal growth is concerned, in hydrothermal synthesis. There is a reference (14), however, this reference appears to be a conference presentation. This is a pretty bold statement to control crystal growth hydrothermally, the authors should expand on how to do this a bit more.
-> This paragraph just introduces the procedure referring Fig.1. But the expression "controlling the nucleation" was not necessarily appropriate. Thus, revised according to the comment and a reference was added.
Line 93 - A reference needs to be added after "Hydrothermal conditions are determined referring the previous researches"
-> Description was not appropriate. Instead of referring the author's previous research, detailed description of the conditions was added.
Line 94 - what was the final pH after the solution was adjusted with ammonia? This is an important experimental detail.
-> Detailed description on how to determine the pH are added. Related material is attached to the reviewer.
Line 102-103 - The authors state that "This means that the small urchins dissolved and then snthesized as bigger urchins." Is there actual evidence of this? This should be described as a proposed theory since hydrothermal synthesis cannot be monitored and viewed in real-time.
-> Description was not appropriate. Actually the discussion is just a hypothesis based on the observation results. Thus, revised according to the comment.
Line 104 - referring to equations 1-6 - The authors should describe this ias a proposed mechanism. If there is evidence that this is certainly the mechanism - text should be added to address that. If they got this mechanism from somewhere else, it should be referenced.
-> The point is that the reaction is not one way but bi-directional as referred. Considering both of this reaction and change in the distance between urchins, the authors considered a hypothesis. Description was added.
Line 141 - In this case the pH was adjusted with HCl, however, the authors do not say what the pH was adjusted to, or how much HCl was added. The authors should say what pH the solution was adjusted to.
-> Description was not appropriate. Revised according to the comment. pH was not adjusted to certain value but mixed with defined quantity in this case.
Line 142- referring to Table 2 - The authors show the chemicals used in this experiment in the table, however, they do not discuss ratios or amounts as they do in Table 1. This important experimental information should be added here.
-> Table 2 was revised according to the comment.
Line 171 - The authors should specify that they are conducting Powder X-ray Diffraction.
-> Revised according to the comment.
Line 195- In the caption for Figure 7, the authors states that the "hydrophilicity was kept for a long time" and there is a reference present. I have two comments here, a "long time" is all relative, therefore, the authors should put a minimum amount of time here that the hydrophilicity was unaffected. Additionally, why is there a reference present here, when the hydrophility length of time is being discussed?
-> Description was not appropriate. Revised according to the comment including the main text.

Reviewer 3 Report
Paper presents two important complementary information for kind readers:
1) how to produce a new sort of materials exhibiting required physico-chemical properties
2) how work these newly obtained systems
The well-treated part 2) of the paper satisfactorily provides experimentally obtained results with a relatively high potentiality in the applications (e.g., gas sensors in the case of ZnO-based systems). The technological part 1) of the presented article needs further elucidation. I'll try to formulate the essential points of this as follows:
a) As known, ZnO exists in two structural modifications: hexagonal wurtzite and cubic zincblende. Why did you study only the first possibility? Stability aspects?
b) Authors applied the seeding method (using ZnO nuclei) to prepare active sites for the nucleation and subsequent growth of required structures. How the size of ZnO nuclei influence the process of self-assembling? How authors controlled nucleation on these ZnO nuclei?
c) What parameters (in particular, adjustable from the outside, such as, e.g., temperature, wettability, velocity) may influence assembly/disassembly process of ZnO clusters?
d) What is the clusters concentration? How this quantity influences the possible transition between different structures?
e) What factors influence the monolayerity of the system?
f) How the velocity of substrate drawing from the suspension affects the final structural properties?
And the basic question:
What is the principal morphology "controller"?
Author Response
a) As known, ZnO exists in two structural modifications: hexagonal wurtzite and cubic zincblende. Why did you study only the first possibility? Stability aspects?
-> The reason is just easiness of the experiments in addition to the stability. The case of cubic zincblende is one of the future problems.
b) Authors applied the seeding method (using ZnO nuclei) to prepare active sites for the nucleation and subsequent growth of required structures. How the size of ZnO nuclei influence the process of self-assembling? How authors controlled nucleation on these ZnO nuclei?
-> In the case of dip coating, the particles size should be smaller than 1 micron meter typically. The authors used the particles of 100 nm in diameter in this case. It was a nominal value of the manufacturer and the authors could not control the size.
c) What parameters (in particular, adjustable from the outside, such as, e.g., temperature, wettability, velocity) may influence assembly/disassembly process of ZnO clusters?
-> The authors showed the effect of the parameters in Ref. 9, but briefly described as following. Aggregation of particles results in irregular assembly. Electrostatic force between particles are important factor, thus, pH was modified based on zeta potential in some cases. Self-assembly process is dynamic and complicated process because at the stage of suspension, the aggregation of particles should be avoided. However, at the last stage, meniscus attraction force between the particles builds close packed structures without any control. Thus, hydrophilicity of the particles surface is essential. Ideally, the water contact angle should smaller than ten degrees.
d) What is the clusters concentration? How this quantity influences the possible transition between different structures?
-> The concentration of the suspension was 1 wt%. With the increase in the concentration, the structure became multi-layered.
e) What factors influence the monolayerity of the system?
-> Concentration of the suspension, drawing-up velocity, and wettability of the substrate affect the number of layers.
f) How the velocity of substrate drawing from the suspension affects the final structural properties?
-> Generally, with the increase in the velocity, the width and number of layer decreases. Related material was attached.
And the basic question:
What is the principal morphology "controller"?
-> Based on the experience, concentration of the suspension and wettability are the principal factors. Related material was attached and the manuscript was revised according to the comments.

Round 2
Reviewer 2 Report
The authors addressed the cornerns sufficiently from my first review of the manuscript.